# Investigation of Forging Metal Specimens of Different Relative Reductions Using Ultrasonic Waves

**DOI:** 10.3390/ma14092406

**Published:** 2021-05-05

**Authors:** Ján Moravec, Peter Bury, František Černobila

**Affiliations:** 1Department of Technological Engineering, Faculty of Mechanical Engineering, University of Žilina, Univerzitná 8215/1, 010 26 Žilina, Slovakia; 2Department of Physics, Faculty of Electrical Engineering and Information Technology, University of Žilina, Univerzitná 8215/1, 010 26 Žilina, Slovakia; peter.bury@uniza.sk (P.B.); frantisek.cernobila@uniza.sk (F.Č.)

**Keywords:** forging metal specimens, ultrasonic investigation, Barkhausen noise, hidden defects

## Abstract

Forgings produced in industry are an irreplaceable basis for subsequent elaborating on machine tools. The quality of the semi-finished product produced by forging is a necessary prerequisite for ensuring the final quality of the final product because the forging can produce some defects. The presented paper is aimed at investigation of selected characteristics of forging steel specimens for various levels of their relative reduction. Ultrasound testing belongs to methods for investigation of structure changes, including defects. Experimental investigation, using both the attenuation and velocity measurements, verify that the reduction of specimens’ material can have an effect on the propagation of ultrasound waves passing through the specimen body. The procedure of steel samples forging corresponds accordingly to the process of their hardening. The increase of toughness after relative reduction of forging in the range of 10–50% is with highest probability caused by the strength matrices development due to the relatively important deformation hardening. It is evident that the deformation hardening is almost the same after every 10% addition of relative reduction. Experiments are supplemented by Barkhausen noise detection and metallographic characteristics of the samples. While differences between the Barkhausen noise values are in principle relatively small and significant differences are only in the values of the position of the envelope, there is maximum coincidence with ultrasonic investigation.

## 1. Introduction

Metal beating is one of the basic forming processes, especially under dynamic loading. Hammering belongs to the field of forging and is the most common operation on impact machines (hammers, drop hammers, spindle presses, etc.). The beating itself takes only a few fractions of a second, which means that the process cannot be considered isothermal. In addition to its practical significance, beating is of fundamental importance for determining the mechanical properties under impact loading. Only the pressure test allows the tool to immediately hit the material. The theory of plastic deformation propagation, taking into account the influence of the deformation rate, the thermal effect, the actual course of the load and the final dimensions, is still under investigation. With sufficiently short samples, the propagation of plastic deformation can be neglected. Only short samples can be used for beating on conventional machines. It is known from practice that the maximum length of loose material that can be broken without deflection is l = 3d. When hammering, the height of the material always decreases. Various tests are applied to test forgings, such as checking the chemical composition of the material, checking the mechanical properties and structure of the forgings. These are commonly used tests in which forging errors can be detected. In particular, the following methods are used in practice to detect hidden forging defects:Assay of electromagnetic waves of different lengths;Test of magnetic and electrical phenomena—capillary tests;Magnetoinductive methods;Magnetographic method;Magnetic powder method;Magnetic fluorescence method;Radiation methods;X-ray defectoscopy;Ultrasound defectoscopy.

In the experimental part, attention was focused on ultrasonic waves and their attenuation.

Quantities characterizing the propagation of high-frequency acoustic waves (ultrasound) in solid materials can be determined from the investigation of dependence of both acoustic wave velocity and attenuation on individual variables describing investigated properties of existing materials. The relation of velocity and attenuation of ultrasonic waves to various properties referring to intrinsic structure of solid material can then enable their study by means of ultrasonic velocity and attenuation measurement. Among these properties, mainly mechanical (elastic, structural, etc.) properties of materials, which directly reflect their intrinsic structure, are considered [1,2].

Using the measurement of acoustic wave velocity that is in proportion to the root of ratio of elastic constants and density of investigated materials, we can receive information connected directly with interatomic forces or density. The ultrasonic wave attenuation very sensitively reflects material intrinsic structure, not only the type of structure but also the changes due to occurrence of defects (dislocations, grains, cracks etc.) Ultrasonic attenuation and velocity in metal materials depend on the method of preparation and further treatment, which can influence grain magnitude as well as the generation of some defects. The attenuation also reflects permanent changes in material arising due to its strain [3]. In the case of material with a grain structure, dislocations, cracks or other defects, the attenuation is caused by the dispersion, the frequency dependence of which depends on the rate of wavelength and mean defect size [4].

The issue is quite extensive and the number of papers devoted to the study of the material characteristics of forgings is large. The most important is [5]. The authors deal with ultrasound control used in non-destructive testing to detect cracks and other defects. It is an important publication and provides a good overview of the issue. Other contributions, such as [6,7], deal with large forged steel parts, testing the phase field by ultrasound in each directional reaction. The CIVA software package (EXTENDE S.A., Massy, France) was used to optimize the phase field. The three test samples described in the source [8] were taken from the material so that the UT indications were in the middle of the cross-section. The samples were then subjected to cyclic loading by applying tensile stresses of different magnitudes and different stress ratios. The experimental investigation in the article [9] confirms the challenges and the current shortcomings in the control of planned industrial components, where such microstructures are desirable in terms of their mechanical properties. Article [10] is an important and significant work, where the interaction between ultrasonic acoustic radiation and the microstructure of duplex stainless steel 2205 was studied. Samples were examined at 780 °C to promote precipitation of intermetallic phases. The UT response in each sample was measured, associated with the respective microstructural properties. Article [11] summarizes the results of research into the effect of sensitivity distribution of double transducer probes, which are often used in non-destructive ultrasonic testing of forgings. The sensitivity distribution measured in two directions, parallel and perpendicular to the separation plane of the dual sensor probe, was tested and analyzed. The approach presented in paper [12] combines the technique of focusing with synthetic aperture (SAFT) to accurately find and quantify small errors. The ultrasound inspection data obtained in the pulsed echo configuration are reconstructed using the synthetic binding application coupling (SAFT) technique. Document [13] investigated the relationship between changes in ultrasonic group and phase velocity with the local properties of a forged and heat-treated large bainitic steel block. From the obtained data it is possible to find out the number and approximate location of small errors. Article [14] describes the development of a new piezoelectric machine operating in a continuous mode. The machine was used to investigate the very high cyclic fatigue (VHCF) properties of the VT3-1 alpha beta aviation titanium alloy. This is produced by two production technologies: forging and extrusion. Extruded titanium alloy has a higher torsional strength of VHCF compared to forging. Ultrasonic testing was used to evaluate the quality of the joints [15]. The basic rising process was chosen to investigate the forging. The good quality of the joint after forging was examined by a penetration test and optical microscopy. A reliable connection between the two materials is important for optimal stress transfer between the materials throughout the component. A special bending test was developed to determine the interfacial strength. The paper [16] describes the problem of the large size of the radial structure and the high mechanical power of the MP1000 miner spindle forging. The low efficiency and reliability of ultrasonic non-destructive tests make it difficult to detect uneven material in the forging process. The numerical finite element method itself, the simulation of the process of detecting ultrasonic errors of the spindle forging, is performed using software. To classify internal defects in the forging, the authors proposed a method [17] based on sample detection of errors obtained in ultrasonic tests. The reliability of the method is more than 75%. The authors also developed a program for the classification of internal defects in relation to the length and their location across the thickness and width of the forging. The pulse compression technique was applied in [18] for ultrasonic inspection of the forging. Currently, (some) authors have applied it in combination with the use of broadband ultrasonic transducers and broadband excitation signals. The methods are extended by applying distance gain size (DGS) analysis to the output pulse compression signal. The assessment and assessment of cracks [19] when deciding on suitability for use require an examination of the location and size of cracks in hazardous areas. An ultrasonic transducer is mainly used for such evaluation, but classical methods cannot accurately evaluate the forging, such as mainly wheel hubs.

It is also possible to make other contributions to the issue. In [20,21,22], the authors deal with the processing of ultrasonic field signals and [21] deals with the problem of velocities in steel blocks, which is very stimulating for the whole area of such research. In addition to these papers, there is also [22], where the authors present an interesting area of pulsed compression ultrasonic technology, where a detailed course of steel inspection is described. The following literature [23,24,25,26,27,28] deals with and presents in detail non-destructive evaluation systems with the application of ultrasound, so it is probably not necessary to discuss these articles in more detail. Article [25] describes the problem of the speed of ultrasonic waves as one of the suitable methods for investigating the properties of materials and forgings by non-destructive methods. The paper [27] focuses on the presented issue of large-scale parts [6,7,13]. In the paper [29], the authors dealt with a wide range of issues of analysis and research of texture issues in metals. The last contribution of this review is [30], which interestingly describes the issue of ultrasound in its application to complex and rugged surfaces of metal parts. The hot deformation behavior and pulse current assisted diffusion bonding (PCDB) behavior of sintered γ-TiAl-based alloy with near gamma microstructure were investigated for fabricating honeycomb structure via isothermal forging—PCDB route. Additionally, the mechanical property and structural morphology demonstrated that the best forging with parameter was forging at 1200 °C with a nominal strain rate of 10^−3^ s^−1^ subsequent to the effects of bonding [31]. In work [32], the influence of weak interface between particles and matrix on mechanical properties of metal matrix–ceramic reinforced composites is studied. Firstly, the samples made of coelectrodeposited Ni-SiC composites with 10% of SiC with poor interface bonding were prepared. Furthermore, the tensile test of samples was performed. The determined Young’s v modulus was equal to 67 ± 8 GPa and the ultimate tensile strength to 230 ± 15 MPa.

In the presented paper, the samples are of simple shapes and it is not necessary to apply complicated procedures for the investigation of the investigated characteristics. The presented contribution is orientated toward analysis of changes of mechanical properties in steel samples that undergo forging with respect to the original non-forging sample. The structure changes of existing material were investigated for various relative reductions of forging deformation using both attenuation frequency dependence and ultrasound velocity measurements. Obtained results were also compared with results obtained by another method on the same samples, specifically by means of the characteristics of the samples using the Barkhausen noise method and metallographic observation of the sample material.

## 2. Ultrasonic Investigation

### 2.1. Experimental Details

Investigated material was chosen from non-operated steel of composition described in Table 1 and, from six sample series corresponding to six different material conditions, original non-forging sample (0) and five forging samples with relative reduction 10% (1), 20% (2), 30% (3), 40% (4) and 50% (5) were prepared. The relative reduction of forging steel samples was calculated using the relation
*Φ* = ln (*h*_0_/*h*) × 100%,(1)
where *h*_0_ is the original height and *h* is the height after forging.

The logarithmic degree of beating for individual samples was assigned according to Table 2.

The samples were subjected to a reduction ranging from 5% to 50%. One sample was uncompressed. The actual dimensions of the compressed samples and their reduction are shown in Table 3.

Samples for the forging were prepared in the cylinder shape with diameter *d* = 13 mm and original height *h* = 12.85 mm. The velocity and attenuation of ultrasound were measured for ultrasonic wave propagated in the direction of cylinder axis and by that in deformation direction, but after grinding and polishing of both cylinder surfaces. The ultrasound pulses of six different frequencies from the range 5–30 MHz were generated by the quartz transducers using the Pulse Modulator and Receiver—MATEC 7700 (Northborough, MA, USA). The same quartz was used for receiving the reflected ultrasonic wave. The attenuation was measured using MATEC Attenuation Recorder 2470 A. All measurements were performed at room temperature. However, the higher frequencies of ultrasound (˃30 MHz) due to the extremely high attenuation could not be used.

### 2.2. Experimental Results and Discussion

The results of velocity measurement for all investigated samples and for deformation directions in cylinder axis are shown in Figure 1.

The velocity measurements were made at frequency of 12.2 MHz with accuracy better then 2 × 10^−3^. It is seen that the value of ultrasound velocity increases with increasing relative reduction up to sample *3* (30%) almost linearly and after that, there is only a small attenuation rise to sample *4* (40%), and then it continues again in the early trend to the last sample *5* (50%). Such behavior, also taking into account the increase of the density with increasing relative reduction, corresponds to the considerable increase of elastic constant. The procedure of steel samples forging corresponds accordingly to the process of their hardening. The increase of the toughness after relative reduction of forging in the range of 10–50% is with highest probability caused by the strength matrices development due to the relatively important deformation hardening. It is evident that the deformation hardening is almost the same after every 10% addition of relative reduction, excepting the mentioned rise between 30% and 40% of reduction, however.

The frequency dependences of ultrasonic attenuation measured for individual samples are illustrated in Figure 2. By contrast to velocity measurements, the development of frequency dependence of attenuation shows at first the increase of attenuation magnitude in the case of first deformed sample *1* (10%) in regards to the non-forging sample but is then followed with its decrease with increasing relative reduction. The decrease of ultrasonic attenuation with increasing relative reduction fully corresponds to velocity measurements and by that to the deformation hardening. Even the small rise of velocity between 30% and 40% of relative reduction corresponds to a small decrease of attenuation between the same samples. The fact that the initial increase of attenuation of sample *1* (10%) does not correspond to results of velocity measurement could be connected with some defect in creation after the first forging. As these defects do not influence the hardening process, they could affect the ultrasonic wave dispersion. Because the frequency dependence of attenuation can be described by the same function, except perhaps for the last one (50%), it practically confirms that the relative reduction caused by forging process at least to 40% does not create additional defects.

The results obtained by measurements of attenuation and velocity correlate considerably with results of authors’ measurements obtained on the same samples, mainly by the Barkhausen noise detection and metallographic characteristics.

## 3. Investigation of Sample Characteristics Using Barkhausen Noise

### 3.1. Experimental Results

Another method used for the analysis of changes of mechanical properties in steel samples that undergo forging with respect to the non-forging sample is the Barkhausen noise method. The basic parameters used in this method were: magnetization frequency 125 Hz, voltage 5 V, serial sensor and Barkhausen noise in frequencies range from 10 to 1000 kHz. The obtained characteristics and dependences of the rollers are shown in Figure 3, Figure 4, Figure 5, Figure 6, Figure 7, Figure 8, Figure 9 and Figure 10.

### 3.2. Evaluation and Discussion of Barkhausen Noise Test

Magnetic parameters do not show a clear trend. Barkhausen noise (MBN) decreases slightly from roller no. 0 to roller no. 2 and consequently the values for the last three rollers no. 3 to 5 are higher compared to roller no. 2, around the value of 345 mV. The position of the envelope maximum (which usually corresponds to mechanical and magnetic hardness) as well as the width of the envelope at half its height (FWHM) also show an ambiguous trend. The differences between the Barkhausen noise values are in principle relatively small. Significant differences are in the values of the position of the envelope maximum. Overall, it can be said that the experimental Barkhausen noise data are essentially unsuitable for the characterization of these samples, or that it is now difficult to interpret these data, as the information from conventional destructive tests is not known.

## 4. Metallographic Observation

The attached pictures of sample microstructures obtained by metallographic microscope (Figure 11) show the surfaces of the investigated samples after their forging with respect to the original non-forging sample. The samples were cut in the longitudinal axis and etched. The surface is magnified 500× to make the structure of the material well observable. The marking of the individual samples is in accordance with the previous one. It is evident that the sample microstructure changes with increasing relative reduction. It can be seen that the samples’ grains, including free space between them, are reduced in a way that corresponds to the increase of their compaction and/or hardening. However, it seems that after relative reduction 40% (e) the sample microstructure does not quite coincide with the described trend of structure changes, but this fact corresponds with ultrasonic investigation, primarily with the ultrasound velocity changes with increasing relative reduction.

In the experimental part, attention was paid preferentially to the shape of transient waves and attenuation. The hardness of the samples remained at an overall low value and is supplemented for a complete view of the problems of the samples used. The hardness values are summarized in Table 4.

## 5. Conclusions

In this contribution, the analysis of changes of mechanical properties in steel samples that undergo forging with respect to the original non-forging sample is presented. Experimental examining of ultrasound is proven to be a suitable and preferred method for determining the desired characteristics. The investigation of Barkhausen characteristics is given as a complementary method. The results are also supplemented by metallographic images of all samples. In conclusion, we can state that methods of ultrasonic attenuation and velocity measurement definitely recorded changes in steel bars that undergo 10–50% relative reduction caused by forging procedure of original steel samples. Experimental results show both velocity increases and attenuation decrease with increasing relative reduction which means that there is then a considerable increase of elastic constant that corresponds accordingly with the process of their hardening. The initial increase of ultrasound attenuation at the lowest value of relative reduction could be connected with some defect creation caused by the forging process at the beginning of the forging procedure and no additional defects, at least to 40% of reduction, are created. While differences between the Barkhausen noise values are in principle relatively small and significant differences are only in the values of the position of the envelope maximum, the pictures of the samples’ microstructure obtained by metallographic microscope coincide well with ultrasonic investigation.

## Figures and Tables

**Figure 1 materials-14-02406-f001:**
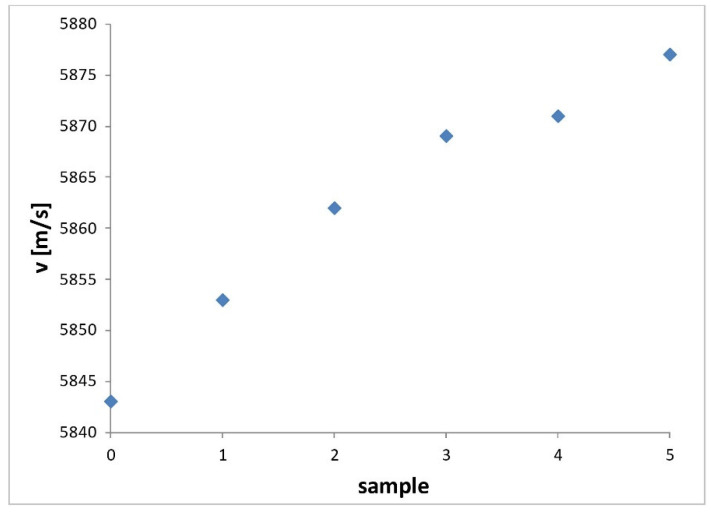
Values of ultrasonic velocity measured at frequency 12.2 MHz for forging samples with relative reduction 10% (1), 20% (2), 30% (3), 40% (4) and 50% (5), including non-forging sample (0).

**Figure 2 materials-14-02406-f002:**
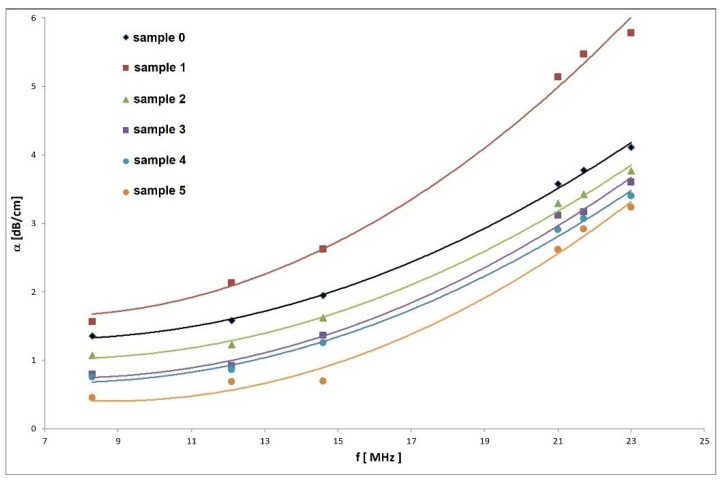
Frequency dependence of ultrasonic attenuation for the samples with different values of relative reduction, 0% (*0*), 10% (*1*), 20% (*2*), 30% (*3*), 40% (4) and 50% (50) measured in frequency range 8.2–23.1 MHz.

**Figure 3 materials-14-02406-f003:**
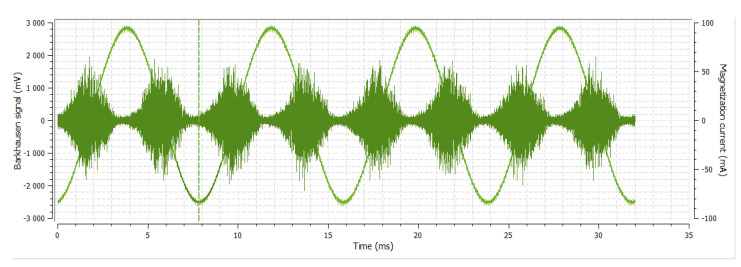
Sample No. 0—Barkhausen noise 339 mV—position of envelope maximum—10.58 a.u.—envelope extension (FWHM—full width at half maximum) 81.2 a.u.

**Figure 4 materials-14-02406-f004:**
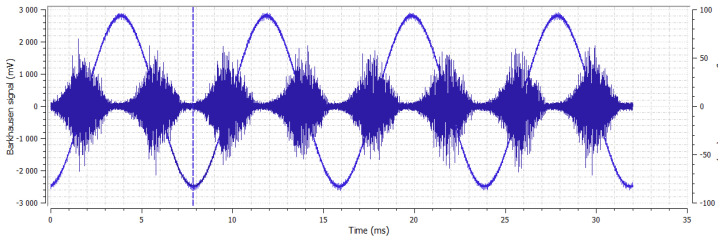
Sample No. 1—Barkhausen noise 334 mV—position of envelope maximum—8 a.u.—envelope extension (FWHM) 77 a.u.

**Figure 5 materials-14-02406-f005:**
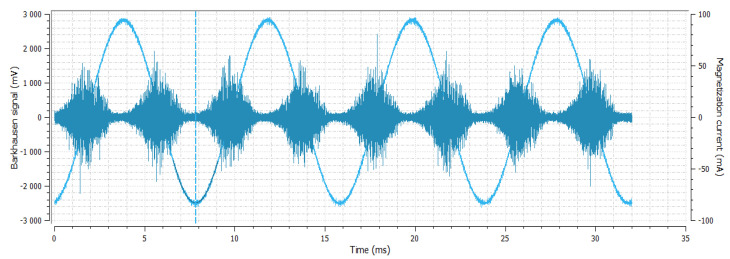
Sample No. 2—Barkhausen noise 300 mV—position of envelope maximum—7.8 a.u.—envelope extension (FWHM) 67 a.u.

**Figure 6 materials-14-02406-f006:**
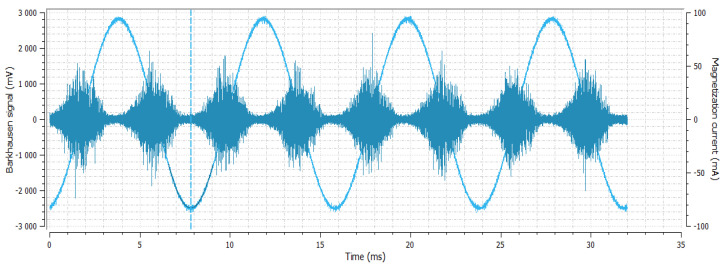
Sample No. 3—Barkhausen noise 344 mV—position of envelope maximum—7.16 a.u.—envelope extension (FWHM) 63 a.u.

**Figure 7 materials-14-02406-f007:**
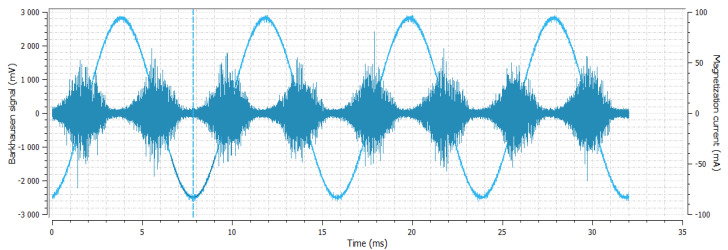
Sample No. 4—Barkhausen noise 374 mV—position of envelope maximum—6 a.u.—envelope extension (FWHM) 70.6 a.u.

**Figure 8 materials-14-02406-f008:**
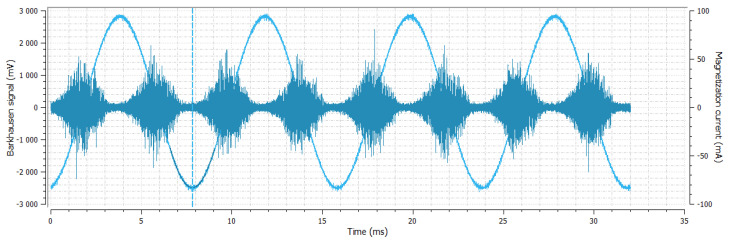
Sample No. 5—Barkhausen noise 340 mV—position of envelope maximum—7.7 a.u.—envelope extension (FWHM) 70.2 a.u.

**Figure 9 materials-14-02406-f009:**
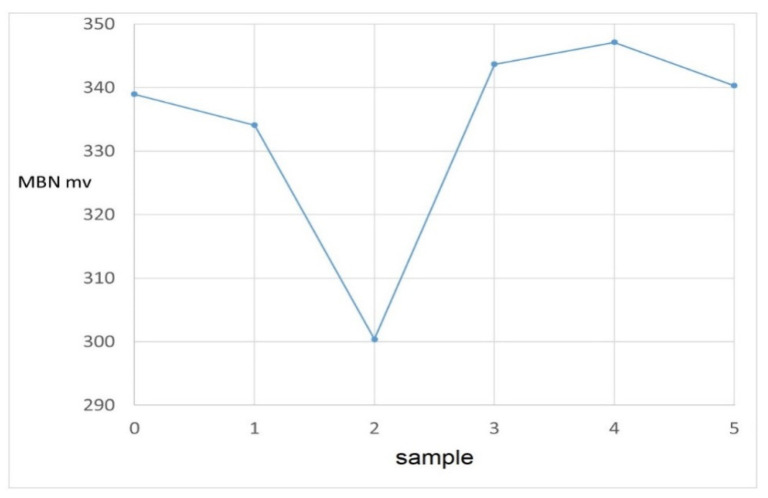
Dependence of rollers and MBN.

**Figure 10 materials-14-02406-f010:**
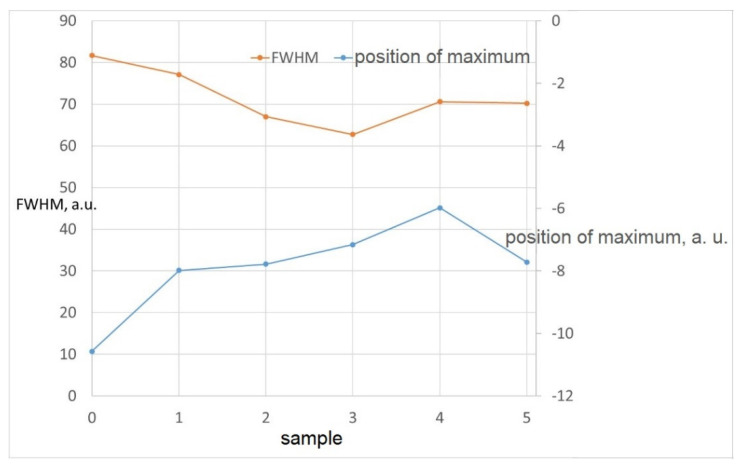
Dependence of rollers and position of maximum and FWHM.

**Figure 11 materials-14-02406-f011:**
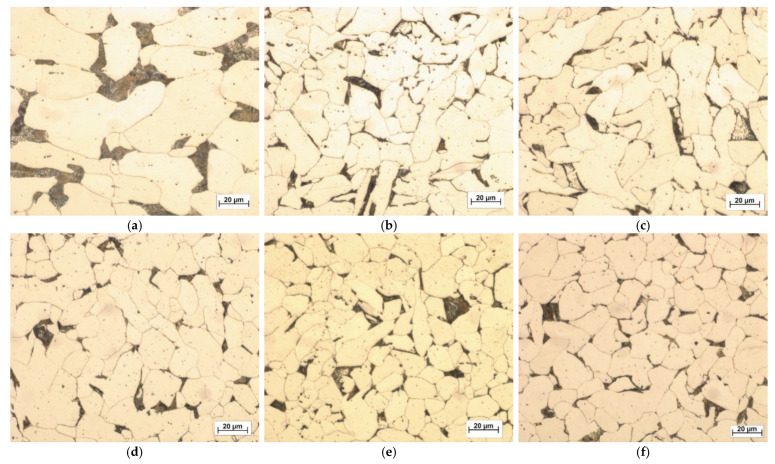
Microstructure of non-forging sample (**a**) and forging samples with relative reduction 10% (**b**), 20% (**c**), 30% (**d**), 40% (**e**) and 50% (**f**).

**Table 1 materials-14-02406-t001:** Chemical impurities in investigated steel in %.

*C* Max.	*Mn* Max.	*Si* Max.	*P* Max.	*S* Max.
0.10	0.45	0.15	0.03	0.025

**Table 2 materials-14-02406-t002:** Logarithmic degree *ϕ* of beating samples.

Sample	0	1	2	3	4	5
Logarithmic degree	0	1.02	1.58	1.094	1.175	1.335

**Table 3 materials-14-02406-t003:** Actual dimensions of rollers height after pressing (mm).

Sample	0	1	2	3	4	5
Height	12.85	12.15	11.18	10.25	8.80	6.75

**Table 4 materials-14-02406-t004:** The hardness of the forging samples.

Sample No.	Test 1 (HV)	Test 2 (HV)	Test 3 (HV)
1	131.6	136.8	133.2
2	140.4	142.8	134.9
3	124.4	128.8	129.4
4	127.1	129.5	126.9
5	131.5	136.2	136.5

## Data Availability

Not applicable.

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
