# Peer review of "Investigation of Forging Metal Specimens of Different Relative Reductions Using Ultrasonic Waves"

_materials, 2021, doi:10.3390/ma14092406_

Round 1

Reviewer 1 Report

Dear Authors,

I have read your paper with pleasure and attention.

In my opinion, the manuscript Investigation of forging metal specimens of different relative reductions using ultrasonic waves presents original research and innovative solution and could be interesting for readers of the MDPI Materials Journal.

The motivation is clear. The object of study, as well as the results, are comprehensively described providing valuable conclusions.

I have no objections to publishing this paper. However, due to the listed below drawbacks, my recommendation is " Accept minor revision". In my opinion, several aspects require clarification. Please make a revision and add some comments and improvements according to the following:

- sentence (line 175): "Parameter: magnetization frequency 125 Hz, volt-175age 5V, serial sensor, Barkhausen noise in frequencies from 10 to 1000kHz." is inelelgant - rewrite, please

- please justify why the frequency 12.2 MHz was chosen and the results for this frequency  are shown in Fig 1? Are the dependencies of ultrasonic velocity measured for all sample for a different frequency similar?

- I believe that in an article in a journal such as Materials MDPI, the information presented in Figs. 3 - 8 should have a different form. The time series of the recorded signals should be exported from the used software and presented independently on the figure, while the characteristic values should be collected and presented in the table included in the text of the article. In my opinion, it is unacceptable to post directly screens of the program window used in analyzes. 

Author Response

Reviewer 1

R – sentence (line 175): "Parameter: magnetization frequency 125 Hz, volt-175age 5V, serial sensor, Barkhausen noise in frequencies from 10 to 1000kHz." is inelelgant - rewrite, please

The sentence (line 175) was corrected:

Another method used for the analysis of changes of mechanical properties in steel samples that undergo to forging with respect to the non-forging sample is the Barkahausen noise method. The basic parameters used in this method were: magnetization frequency 125 Hz, voltage 5 V, serial sensor, and Barkhausen noise in frequencies range from 10 to 1000 kHz.

R – please justify why the frequency 12.2 MHz was chosen and the results for this frequency  are shown in Fig 1? Are the dependencies of ultrasonic velocity measured for all sample for a different frequency similar?

The frequency 12.2 MHz was due to the fact that both the acoustic signal amplitude and the number of acoustic echos, responses after multiplied reflections, were largest and the accuracy of velocity measurement could be highest. No other frequency was used for velocity measurement.

R I believe that in an article in a journal such as Materials MDPI, the information presented in Figs. 3 - 8 should have a different form. The time series of the recorded signals should be exported from the used software and presented independently on the figure, while the characteristic values should be collected and presented in the table included in the text of the article. In my opinion, it is unacceptable to post directly screens of the program window used in analyzes.

Figs. 3-8 were corrected and the screenshot was removed.

Reviewer 2 Report

The article presents the results of a study of forging metal specimens using ultrasound and Barkhausen noise. The article may be of some interest, but as it stands, it is unacceptable for publication, since it requires significant correction.

An abstract is a collection of well-known truths and general words. The abstract should present key research findings in a concise and focused manner. In its current form, the abstract is unacceptable.

The English language requires significant improvement (I recommend professional proofreading or at least correction by a native speaker).

The introduction should not only provide a literary review, but also justify the choice of the direction of research, as well as the novelty and significance of the planned research and their results. That is, a description (short and focused) of the level of research achieved and the shortcomings of the available methods in contrast to the method proposed by the authors.

Fig. 1 and 2. It is advisable to give a decoding of the designations of the axes - directly on the graphs or at least in the figure caption. This makes the graphs easier to understand.

Fig. 3-8. First, I see no point in showing a full screenshot (including File, Save, etc.). It is enough to present only the graphs themselves, removing unnecessary garbage. Secondly, I would recommend combining Fig. 3-8 in one Fig. (a, b, ...).

I also recommend combining Fig. 9 and 10.

Fig. 11 - what's the point in these images? There is practically no description for them. How do these images relate to previous results? Fig. 11 must be described and analyzed in detail (including in connection with previous results) or deleted.

The connection between Tab. 4 and Fig. 12 with the results of ultrasound and Barkhausen noise tests. What is the connection, what is the impact? This should be described clearly and in sufficient detail.

Fig. 9, 10 and 12. The values are connected by straight lines. Are there intermediate results? Why straight (not curved) lines? Why are the error bars not indicated?

Tab. 4 and Fig. 12 represent the same information. What's the point of this?

Author Response

R – An abstract is a collection of well-known truths and general words. The abstract should present key research findings in a concise and focused manner. In its current form, the abstract is unacceptable.

The Abstract was corrected in in regards to obtained results.

R – The introduction should not only provide a literary review, but also justify the choice of the direction of research, as well as the novelty and significance of the planned research and their results. That is, a description (short and focused) of the level of research achieved and the shortcomings of the available methods in contrast to the method proposed by the authors.

The Introduction was extended and corrected.

R – Fig. 1 and 2. It is advisable to give a decoding of the designations of the axes - directly on the graphs or at least in the figure caption. This makes the graphs easier to understand

The figure captions of Fig. 1 and Fig. 2 were extended:

Figure 1. Values of ultrasonic velocity measured at frequency 12.2 MHz for forging samples with relative reduction 10% (1), 20% (2) 30% (3), 40% (4) and 50% (5), including non-forging sample (0).

Figure 2. Frequency dependence of ultrasonic attenuation for the samples with different value of relative reduction, 0% (0), 10% (1), 20% (2) 30% (3) 40% (4) and 50% (50) measured in frequency range 8.2 – 23.1 MHz.

R Fig. 3-8. First, I see no point in showing a full screenshot (including File, Save, etc.). It is enough to present only the graphs themselves, removing unnecessary garbage. Secondly, I would recommend combining Fig. 3-8 in one Fig. (a, b, ...).

The screenshot was removed and Figs. 3-8 present only graphs.

R – Fig. 11 - what's the point in these images? There is practically no description for them. How do these images relate to previous results? Fig. 11 must be described and analyzed in detail (including in connection with previous results) or deleted.

The description of Fig. 11 was completed. Evident compaction of the sample material can be seen in the published images. Differences in different reductions are clearly observable.

R – The connection between Tab. 4 and Fig. 12 with the results of ultrasound and Barkhausen noise tests. What is the connection, what is the impact? This should be described clearly and in sufficient detail.

We agree, Fig. 12 presented the same results as Tab. 4 and was removed.

R – Fig. 9, 10 and 12. The values are connected by straight lines. Are there intermediate results? Why straight (not curved) lines? Why are the error bars not indicated?

The straight lines do not represent any function, they only enable to follow measured changes better.

R – Tab. 4 and Fig. 12 represent the same information. What's the point of this?

Fig. 12 presented the same results as Tab. 4 and was removed.

Reviewer 3 Report

  1. Summary, strengths, weaknesses, overall contribution

In the paper the Authors investigate  ultrasound  characteristics  of reduced, forged specimens for various levels of their reduction. The objective is to verify, whether the reduction of specimens’ material can have an effect on the propagation of ultrasound waves passing through the specimen body. The weakness of the paper is poor quality of the writing as well as a lack of discussion. Overall contribution is unclearly communicated in the paper, however after major revision, it can be interesting for the ultrasound or non-destructive methods communities.

What is also important, the Authors presents some results but do not discuss them in detail. The part named “discussion” should be almost entirely moved to the introduction part.

English must be improved. I am not a native speaker, but still I have noticed many spelling errors (e.g. “rice instead of rise”)

  1. Major comments

The paper may be reconsidered for publication  if the authors will refer to the following remarks and do the necessary corrections, which would significantly improve the paper:

B1.  In Introduction, there is a lack of paragraph where the paper itself is described as well as the aim of the research is not clearly and directly given.  Additionally, the Authors mentioned the problem of interfacial bonding strength, however, it is not discussed enough. Authors may refer to the following papers (DOI): 10.1016/j.intermet.2018.05.008; 10.1016/j.compstruct.2018.06.071;

B2. Figure 1 – it would be better to plot velocity vs i.e. reduction instead of velocity vs sample number. The same comment applies to Figure 2.

B3.  Figures 3-8 should be significantly improved. It is not necessary to show the Windows menu bars etc. Furthermore, the Authors should reconsider the idea behind the images. Maybe it would be better to plot the results on one image? What is more, Figures 3 to 8 present exactly the same screen shot.

B4.  There is no discussion of the results in the publication. Most of the part named “discussion” should be moved to the introduction. The part about hardness measurement should be moved to the “results” part. The Authors should comment their results, compare them with other works, discuss the novelty, the importance for the community etc.

  1. Minor comments

C1. Scales in Fig. 11 are cut.

C2. In Fig. 12 and 10 it should be indicated that the lines which connect the points are so called “guide for an eye”. These are not fittings. Furthermore, it would be worth to indicate the measurement errors in the figures.

Author Response

B1 - In Introduction, there is a lack of paragraph where the paper itself is described as well as the aim of the research is not clearly and directly given.  Additionally, the Authors mentioned the problem of interfacial bonding strength, however, it is not discussed enough. Authors may refer to the following papers (DOI): 10.1016/j.intermet.2018.05.008; 10.1016/j.compstruct.2018.06.071;

Results of the first recommended papers were included in the contribution, second one could not be found.

B2 – Figure 1 – it would be better to plot velocity vs i.e. reduction instead of velocity vs sample number. The same comment applies to Figure 2.

Although Figure 1 was not modified, its description was completed according reviewer recommendation:

Figure 1. Values of ultrasonic velocity measured at frequency 12.2 MHz for forging samples with relative reduction 10% (1), 20% (2) 30% (3), 40% (4) and 50% (5), including non-forging sample (0).

Figure 2. Frequency dependence of ultrasonic attenuation for the samples with different value of relative reduction, 0% (0), 10% (1), 20% (2) 30% (3) 40% (4) and 50% (50) measured in frequency range 8.2 – 23.1 MHz.

B3 – Figures 3-8 should be significantly improved. It is not necessary to show the Windows menu bars etc. Furthermore, the Authors should reconsider the idea behind the images. Maybe it would be better to plot the results on one image? What is more, Figures 3 to 8 present exactly the same screen shot.

Figs. 3-8 were corrected and the screenshot was removed.

B4 – There is no discussion of the results in the publication. Most of the part named “discussion” should be moved to the introduction. The part about hardness measurement should be moved to the “results” part. The Authors should comment their results, compare them with other works, discuss the novelty, the importance for the community etc.

The discussion is presented just after results presented obtained by individual methods. The comparison of the results should be done in Conclusion. The part about the hardness measurement was moved to Introduction.

C1 – Scales in Fig. 11 are cut.

Scales in Fig. 11 is ok.

C2 – In Fig. 12 and 10 it should be indicated that the lines which connect the points are so called “guide for an eye”. These are not fittings. Furthermore, it would be worth to indicate the measurement errors in the figures.

We agree, the straight lines do not represent any function, they only enable to follow measured changes better.

Round 2

Reviewer 2 Report

I recommend shortening the abstract, focusing on the main results and conclusions of the article.

Descriptions for Fig. 11 absolutely not enough "Differences in different (English needs to be improved!) Reductions are clearly observable" - what exactly is "clearly observable" - what are the signs, what are the reasons, etc. Fig. takes half a page, consists of eight images and the entire description for it consists of "Evident compaction of the sample material can be seen ..." - but what exactly "can be seen ..." - what are the EXACTLY differences? The article should not contain riddles and puzzles for the reader ("find 12 differences between these pictures"). Authors must clearly describe the differences by CONFIRMING this description with an appropriate image.

Author Response

Dear Reviewer,

we accepted all recommendations required by Reviewer 2. We reduced Abstract and modified the description of Fig. 11 in accordance with his recommendations. We believe that our contribution should be now in convenient form.

Kindest regards    Jan Moravec

Reviewer 3 Report

The paper has been significantly improved. It can be published in the current form.

The paper which I have mentioned in the previous report, which the Authors have not found, can be fund here:

https://doi.org/10.1016/j.compstruct.2018.06.071

Author Response

The post you reqeuested to add i quitte. Contribution (32).